# IIoT Low-Cost ZigBee-Based WSN Implementation for Enhanced Production Efficiency in a Solar Protection Curtains Manufacturing Workshop [note 1]

**DOI:** 10.3390/s24020712

**Published:** 2024-01-22

**Authors:** Hicham Klaina, Imanol Picallo, Peio Lopez-Iturri, Aitor Biurrun, Ana V. Alejos, Leyre Azpilicueta, Abián B. Socorro-Leránoz, Francisco Falcone

**Affiliations:** 1Electric, Electronic and Communication Engineering Department, Public University of Navarre, 31006 Pamplona, Spain; hicham.klaina@unavarra.es (H.K.); imanol.picallo@unavarra.es (I.P.); peio.lopez@unavarra.es (P.L.-I.); biurrun.110701@e.unavarra.es (A.B.); leyre.azpilicueta@unavarra.es (L.A.); ab.socorro@unavarra.es (A.B.S.-L.); 2Institute of Smart Cities, Public University of Navarre, 31006 Pamplona, Spain; 3Signal and Communications Theory Department, University of Vigo, 36310 Vigo, Spain; analejos@uvigo.es; 4School of Engineering and Sciences, Tecnologico de Monterrey, Monterrey 64849, Nuevo León, Mexico

**Keywords:** Industry 4.0, industrial internet of things, wireless sensor networks, manufacturing process, automation, 3D ray launching, wireless channel

## Abstract

Nowadays, the Industry 4.0 concept and the Industrial Internet of Things (IIoT) are considered essential for the implementation of automated manufacturing processes across various industrial settings. In this regard, wireless sensor networks (WSN) are crucial due to their inherent mobility, easy deployment and maintenance, scalability, and low power consumption, among other benefits. In this context, the presented paper proposes an optimized and low-cost WSN based on ZigBee communication technology for the monitoring of a real manufacturing facility. The company designs and manufactures solar protection curtains and aims to integrate the deployed WSN into the Enterprise Resource Planning (ERP) system in order to optimize their production processes and enhance production efficiency and cost estimation capabilities. To achieve this, radio propagation measurements and 3D ray launching simulations were conducted to characterize the wireless channel behavior and facilitate the development of an optimized WSN system that can operate in the complex industrial environment presented and validated through on-site wireless channel measurements, as well as interference analysis. Then, a low-cost WSN was implemented and deployed to acquire real-time data from different machinery and workstations, which will be integrated into the ERP system. Multiple data streams have been collected and processed from the shop floor of the factory by means of the prototype wireless nodes implemented. This integration will enable the company to optimize its production processes, fabricate products more efficiently, and enhance its cost estimation capabilities. Moreover, the proposed system provides a scalable platform, enabling the integration of new sensors as well as information processing capabilities.

## 1. Introduction

In this modern era of technology, Industry 4.0 will allow companies to implement innovative manufacturing techniques by acquiring real-time data using a large number of interconnected smart devices, thanks mainly to the Industrial Internet of Things (IIoT) [1]. In most cases, WSNs are employed for the manufacturing process due to their monitoring capacity, deployment flexibility, and scalability, which leads to an optimization of operating costs. However, the deployment of wireless communication systems, and WSNs in particular, within industrial environments presents specific difficulties and challenges, due mainly to the electromagnetic noise caused by machinery, the intrinsic harsh industrial environment (in many cases with many metallic objects and structures), and the dynamism of the wireless channel, which, added to the present strong multipath propagation, make this type of environment very difficult to analyze in terms of radio propagation planning. In order to tackle these challenges, it is necessary to carry out accurate radio-planning analyses of the wireless channel [2].

The literature presents several studies where different wireless communication technologies and aspects are analyzed within industrial environments. Thus, a survey analyzes WSN reliability in industrial automation and control systems in order to replace wired solutions in factory applications [3]. In [4], another survey analyzes IoT applications in smart logistics, focusing on the challenges and key technologies involved, such as the WSN. In [5], a wireless time-synchronous network that allows machine-to-machine (M2M) data sharing is presented. References [6,7] focus on communication among nodes and real-time data-gathering challenges in industrial environments. In [8], the performance of Long-Range Wide Area Network (LoRaWAN) technology for energy harvesting is explored in industrial automation, including a node lifetime analysis and an assessment of battery replacement. Reference [9] presents a battery-less industrial monitoring system that uses LoRaWAN technology to improve operational reliability in an electrorefinery. In [10], Narrow-Band IoT (NB-IoT) technology is used in order to enhance the digitalization of the oil and gas industry by employing industrial waste heat. Reference [11] presents a rail monitoring system that utilizes IoT sensors and edge computing for real-time data collection, obtaining the train speed and the number of carriages. References [12,13] introduce smart systems for wastewater monitoring using an IoT real-time algorithm to detect and locate harmful wastewater discharges. In [14], the creation and testing of a garbage bin-level monitoring system that operates with LoRaWAN technology and GPS units for recording the location of the trash bin are presented. In [15], self-sustainability and long-range operation Bluetooth sensor node is presented in order to analyze and monitor the aerodynamics of wind turbines. Finally, reference [16] presents the tracking of forklifts in an industrial environment using measurements with Ultra-Wide Band (UWB) technology and simulations with a location algorithm.

Following this trend, this work, which is an extension of a conference proceedings [17], proposes the deployment and optimization of a ZigBee-based WSN within the facilities of the Galeo Enrollables Company, located in Navarre (Spain). The company designs and manufactures technical and solar protection curtains. Due to the current state of lack of control over manufacturing times and administrative tasks for many of the machinery, the company wants to integrate the ad-hoc design of WSN into the Enterprise Resource Planning (ERP) system in order to carry out effective management of the entire manufacturing plant with the aim of controlling and optimizing all their resources and processes. In this sense, the proposed and deployed WSN acquires real-time data from different machinery and workstations within the workshop in order to supply it to the ERP system. The distribution of workstations and requirements within the factory are considered in order to account for the distribution of the wireless transceivers, as well as the number and type of sensors/actuators to be included in the prototype nodes to be developed. Radio planning analysis is performed based on in-house deterministic wireless channel estimation and validated through site measurements in order to assess the selection of the wireless communication protocol. ZigBee technology has been chosen for the wireless connectivity of the sensor nodes due to its intrinsic characteristics, which adapt perfectly to the application within this environment: the very high number of nodes that can be connected to the network (up to 64,000), making it easily scalable; the mesh topology and the dynamic autoconfiguration of the network, which provide robust routing for the sent packets and protection against interference and node failures; and the higher duty cycle compared with other low-power wireless technologies (such as LoRaWAN), which provides the possibility of sending a higher number of packets if required. It is worth noting that the presented methodology could be applied to any wireless communication technology, such as LoRa, NB-IoT, Bluetooth, or ESP-Mesh [18,19,20,21], for monitoring tasks, as long as the requirements of the application are satisfied, and considering that the radio planning study will need to be adapted to each technology’s characteristics.

The paper is organized as follows: Section 2 describes the workshop of Galeo Enrollables Company, the scenario where the proposed system was implemented, as well as its radio characterization prior to wireless communication system deployment. Section 3 presents the proposed wireless sensor network deployment. Results obtained under real conditions are presented and discussed in Section 4. Finally, conclusions are presented in Section 5.

## 2. Curtain Workshop Scenario

### 2.1. Description of the Workshop

The Galeo Enrollables Company, founded in 2008 and located in Navarra, Spain, is a company that designs, innovates, and manufactures technical and solar protection curtains for different types of windows and glass enclosures. With a wide variety of products manufactured, including roller blinds, vertical curtains, Japanese panels, pleated curtains, night and day blinds, nautical curtains, outdoor curtains, and more, the company requires very specific machinery. To meet this need, the factory workshop contains a range of automatic and semiautomatic cutters, welding machines, and automatic machinery slats manufacturing based on crush cutting, laser, ultrasounds and blades, and thermal and ultrasound-based welding machines, among others. Figure 1 shows several views of the main zone (i.e., the workshop) of the factory.

In order to optimize the production process and enhance the cost estimation of its products, Galeo Enrollables Company is currently in the process of integrating a new Enterprise Resource Planning (ERP) system. The system will enable the company to monitor and control the manufacturing cycle time, warehouse stocks, and administrative tasks in real time, leading to the determination of the operating costs of the company. To achieve this, the company is planning to deploy a wireless sensor system to monitor the workshop machinery. This real-time data will be integrated into the new ERP system, enabling the company to optimize its production process, fabricate products more efficiently, and enhance its cost estimation capabilities. Additionally, this will enable the daily planning of production tasks to be automated, increasing efficiency across the board.

### 2.2. Radio Analysis of the Workshop

In preparation for the deployment of the WSN within the Galeo Enrollables Company’s workshop, radio planning measurements were taken in order to study the viability of the proposed ZigBee-based deployment in such a scenario. Specifically, the aim of these initial measurements was to detect potential radio interferences and their sources, such as specific machinery (i.e., laser or soldering machines) or present wireless communication systems operating at interfering frequency bands. The measurements were carried out with the aid of a portable spectrum analyzer on the RF band of 2.4–2.5 GHz, as the WSN to be deployed is based on the 2.4 GHz ZigBee communication technology. No interfering signals produced by the machinery were detected, but the Wi-Fi access points present at the workshop produced RF signals/noise at the desired band, as can be seen in Figure 2. However, the band close to 2.5 GHz was still available, so high-frequency ZigBee channels can be used to avoid potential interference. Additionally, the ZigBee protocol can automatically choose the best operation frequency channel (which is narrower than WiFi channels: 3 MHz vs. 20 MHz) depending on the scenario conditions (e.g., if the channel is free of interferences or not), and the protocol also provides three retransmissions for each lost packet by means of ACKs, minimizing the harmful effect of wireless interferences.

Apart from the interference analysis, it is essential to note that industrial environments exhibit very specific characteristics in terms of radio propagation phenomena. Reflection, diffraction, and scattering produced by different structures and obstacles such as walls, machinery, production lines, shelves, highly reflective surfaces (i.e., big metallic objects), the presence of human beings, etc. result in relevant multipath propagation and, therefore, in a very complex environment in terms of radio propagation [22].

In this context, wireless channel characterization is compulsory before deploying a wireless communication system in this type of environment. For that purpose, an in-house-developed 3D ray launching technique was used to characterize the wireless channel in the presented factory workshop scenario. This technique provides a better understanding of the wireless channel’s behavior and facilitates the deployment of an optimized WSN. By ensuring that the wireless communication system is characterized correctly, the factory will be better equipped to ensure the success of its WSN deployment, leading to a more efficient and productive manufacturing process.

### 2.3. 3D Ray Launching Simulation Software

The 3D ray launching (3D-RL) code used in this study is based on the principles of geometric optics and the uniform theory of diffraction, which involves launching rays from predefined sources with a given angular resolution in both horizontal and vertical planes. The simulation considers all the objects and frequency dispersive material properties, such as the conductivity and relative permittivity of elements in the scenario, including machinery and walls. Diffraction and diffuse scattering are also considered in the simulation. To reduce computational complexity, the simulation code has been optimized to include hybrid simulation options, such as neural network-based interpolators, diffraction estimation based on the electromagnetic diffusion equation, and collaborative filtering database extraction. A detailed description of the code can be found in [23], and it has been tested and validated in various use cases and applications, including industrial environments [24].

The factory workshop scenario created for the 3D-RL simulation has dimensions of 31 m in length, 34 m in width, and a height of 6 m, resulting in an overall volume of approximately 6324 m^3^. Figure 3 shows a view of the complete schematic factory model developed for the simulations. All elements within the workshop, including machinery, metallic elements, and walls, are considered for the analysis of their impact on radio propagation. Table 1 presents the main material properties of all the existing elements within the presented scenario, which are taken from [25].

Once the scenario has been created, a comprehensive series of radio frequency measurements have been carried out to validate the deterministic estimations provided by the 3D Ray Launching simulator. These measurements were taken for the ZigBee frequency band of 2.4 GHz. In order to ensure a fair validation, two different measurement scenarios have been proposed, where different propagation conditions within the workshop have been included and analyzed, such as line of sight (LoS) and nonline of sight (NLoS) conditions. These two measured cases are presented in Figure 4. In the first scenario (Figure 4a), the location of the transmitter (height of 1.75 m), labeled “Tx” in the figure, was chosen to enable a thorough investigation of the RF power level in different areas under both line of sight (LoS) and nonline of sight (NLoS) conditions. The measurement points, represented by numbered red circles, have been strategically distributed covering the entire area of the workshop, being located at different heights on machinery, shelves, and aisles (in aisles at 1.2 m height). This distribution provides valuable information on the radio propagation for the different wireless links present in the workshop. In the second scenario, the study has been designed to evaluate wireless propagation for a linear path, as can be seen in Figure 4b. This path corresponds to an aisle of the workshop, and the aim of this case study is to obtain the RF behavior for a continuous LoS condition. To this end, the measurement points were distributed along this linear path (represented by the white arrow in the figure). The height for the transmitter and receiver in this scenario is 1.2 m, and the distance between the two measurement points is 1 m.

For the measurements, a voltage-controlled oscillator (VCO) has been used to generate the RF signal at 2.4 GHz. The transmitted power at 2.4 GHz was 8.2 dBm, which falls into the typical range of XBee ZigBee motes employed in the final solution. The received RF power level has been measured by means of a portable spectrum analyzer (FSH20 from Rohde and Schwarz, Munich, Germany). Both the transmitter (i.e., the VCO) and the spectrum analyzer have been paired with the Omni-directional Broadband Antenna OmniLOG^®^ 30800 (Gewerbegebiet Aaronia AG II, Strickscheid, Germany), which presents a gain of −6 dBi at the 2.4 GHz band. Figure 5 presents the setup for the measurements.

The simulation of the two case studies presented in Figure 4 has been performed applying the same conditions and parameters used in the empirical experiments, i.e., the same antenna type, transmitted power level, antenna gains, operating frequency, transmitter locations, etc. Table 2 shows a summary of the main parameters configured for the 3D-RL simulations. The simulation tool considers electromagnetic propagation phenomena such as diffraction, refraction, and reflection (the maximum number of permitted reflections of a ray is set by the user; in this case, it has been set to 6, as given by the convergence analysis described in [26]). It is worth noting that the employed 3D-RL simulator provides results for the complete volume of the scenario under analysis, enabling the analysis of any potential device location at any given height if required.

Figure 6 presents the comparison between measured and simulated results in terms of received RF power level. Figure 6a shows the results corresponding to 16 measurement points from Figure 4a (scenario 1), while Figure 6b shows the results corresponding to the linear path presented in Figure 4b (scenario 2). As can be seen, the results show a good match, validating the simulator for further radio analysis tasks within the workshop under analysis. Additionally, Table 3 presents the obtained mean error, the variance, and the standard deviation for the two analyzed scenarios. It is worth noting that the proposed methodology can be extended to any required frequency range, enabling the consideration of other wireless communication standards as a function of coverage/capacity requirements, such as PLMN-based IoT connectivity (e.g., 4G NB-IoT/LTE Cat. M, 5G D2D connectivity mainly within the 5G NR FR1 band, although extendable towards 5G NR FR2, etc.), evolutions in LPWAN, or updates in 802.11 IoT connectivity (e.g., 802.11 ah WiFi Ha-Low).

## 3. Wireless Sensor System Deployment

### 3.1. Radio Planning

Once the simulation tool has been validated and prior to the deployment of the WSN within the workshop, a brief 3D-RL-based radio planning study has been performed to gain insight regarding the feasibility of deploying the desired and proposed ZigBee-based WSN within the curtain factory workshop. The simulation parameters are the same as those presented in the previous section. The aim of this analysis is to help understand the radio propagation behavior and the connectivity limitations within the workshop.

First, the PDPs (power delay profile) depicted in Figure 7 show the propagation complexity of the scenario under analysis. These estimated time domain results provide a clear view of the multipath propagation complexity at each point within the scenario. Four different locations are presented in the figure as examples, corresponding to the measurement points 3, 10, 13, and 15 (at a height of 1.3 m) presented in Figure 4a. Transmitter 1 results correspond to transmitter locations shown in scenario 1 (Figure 4a), and Transmitter 2 results correspond to scenario 2 (Figure 4b). It has to be mentioned that multipath components with less than −100 dBm are not presented in the graphs, since this is the sensitivity level of the ZigBee motes employed in this work.

As expected, the environment is very rich in terms of multipath propagation due to the presence of several metallic scatterers and different objects, typical in industrial environments. Moreover, in this case, the LoS and NLoS conditions could have a significant effect on the wireless link performance.

In order to analyze the feasibility of the deployment of ZigBee wireless nodes and the quality of the wireless link between them, RF power distribution results have been obtained. These results have been obtained for the whole 3D volume of the workshop for both scenario 1 and scenario 2. Obtaining results for the whole volume of a scenario leads to a complete radio planning analysis since the radio links can be analyzed even if the transmitters and receivers are deployed at different heights, where the RF power distribution will vary. As an example, Figure 8 shows the RF power level distribution for three different heights (i.e., 1 m, 2 m, and 3 m from the floor) for scenario 2. Note that the transmitter (Tx in the figure) is located at height = 1.3, but the black dots representing it are drawn in every map for a better understanding of the setup. It can be noted that, when the height is increased, the shadowing effects due to obstacles (mainly machinery) tend to disappear, since the machinery distributed throughout the workshop presents an overall height of 1.2 m.

Returning to the cases under analysis, Figure 9 shows the obtained estimations for both scenario 1 and scenario 2. Specifically, 2D planes at a height of 1.3 m are presented, where the main RF power drops are caused by shadowing effects and NLoS situations due to walls and machinery. In this way, it can be seen how the detected RF power in some areas falls near and under the −100 dBm sensitivity threshold of ZigBee motes (dark blue points in the coverage map). This means that in some specific cases, the wireless communication between two ZigBee motes deployed within the scenario could fail. In order to gain insight into this matter, Figure 10 shows the areas where the RF power level is higher (green zones) and lower (red zones) than the sensitivity level of the motes. As can be seen, most of the planes’ areas are green, which means that receivers deployed there will comply with the required sensitivity level for the simulated specific Tx locations. On the other hand, red zones or points show where the deployment of a receiver will not comply with the sensitivity of the motes, leading to failed connectivity.

Summarizing, the wireless connectivity between two motes employing the chosen wireless communication technology (i.e., ZigBee) operating within the workshop under study presents good results, although in some specific cases, successful connectivity could not be obtained. However, this possible issue was one of the reasons why ZigBee technology was chosen for this WSN deployment. The ZigBee network can be deployed in mesh topology, which in practice means that wireless communication will be performed by routing the packets among the deployed nodes until they reach the ZigBee gateway (i.e., coordinator). In fact, the mesh topology provided by the ZigBee protocol is what has been employed in the deployment shown in the next section, which provides robustness to the communications since it is managed automatically and dynamically (depending on the changing conditions of the wireless links between nodes) by the protocol. Thus, taking a look at the obtained results, the inclusion of each extra node will improve the connectivity, covering the entire workshop easily, corroborating that, in principle, the presented solution is adequate for the environment under analysis.

### 3.2. WSN Deployment

After analyzing the radio characteristics of the scenario, this section presents the deployment of the ZigBee-based WSN within the workshop. The results for both scenarios, as presented in the previous section, suggest that positioning the network coordinator in the middle of the workshop would improve the RF propagation, particularly avoiding the walls in the corners, i.e., avoiding NLoS situations. Thus reducing the low coverage areas and improving end-to-end communication within the entire scenario. The proposed locations of the coordinator (marked as ‘C’) and the five workstations monitored by sensor nodes are illustrated in Figure 11. Depending on the characteristics and requirements of each workstation, the sensor nodes’ components differ, but in general, they consist of an Arduino UNO board, an IO Shield expansion, digital push buttons, a matrix keypad, KY-032 and KY-033 passive infrared (PIR) sensors from Joy-IT, real-time Clock (RTC) modules for synchronization, and an XBee s2c module for the ZigBee wireless communications, operating at its higher transmitted power level (i.e., 8 dBm). The difference between KY-032 and KY-033 modules is the detection distance. The KY-032 module can detect obstacles 40 cm away from the sensor, while the KY-033 module has a maximum detection range of less than 10 cm. Figure 12 presents schematically the implemented sensor nodes and coordinator/gateway, as well as a real sensor node and its encapsulation, ready to be deployed.

Five workstations have been selected for real-time monitoring due to their importance in the fabrication process of the solar curtains. Figure 13 shows the deployed sensor nodes on the five workstations. As can be seen, different number of nodes have been deployed depending on the parameters to be monitored on each workstation. The roles of the sensor nodes and the goals behind its implementation within each workstation are described below:Workstation 1: The Knife cutting machine (see Figure 13a). In this node, both KY-033 and KY-032 sensors have been used. IR obstacle sensor KY-033 is used to detect the passage of the cutting blade. With this, it detects the time that elapses between the different cuts. Moreover, IR obstacle sensor KY-032 detects when the fabric roll is deposited. This results in the time that elapses in the search for tissue. The green pushbutton is used to terminate each product that has been made and the red pushbutton is used to collect the time of searching for tissue scraps. Finally, the Keypad is used to enter the identification of the operator, the number of scraps that have been taken and the production order to be made. This node has been implemented with the aim of collecting the following data: Production order that is being carried out; operator identification; time at which production times are being carried out (preparation and search time for the fabric, time between each cut of fabric, collection of the semifinished product, total time).Workstation 2: Laser cutting machine (see Figure 13b). The IR obstacle sensor KY-032 detects when the fabric roll is deposited. This results in the time that elapses in the search for tissue. The green pushbutton is used to terminate each product that has been made and the red pushbutton is used to collect the time of searching for tissue scraps. Finally, the 3 × 4 Matrix Keypad is used to enter the identification of the operator, the number of scraps that have been taken and the production order to be made. When the machine is with the laser on (cutting), the laser value is obtained directly from the PLC of the machine. The objective of the node implementation is to collect the following data: Production order that is being made; operator identification; time at which production times are being carried out (preparation and search time for the fabric, extension of the fabric, fabric cutting time, collection of the semifinished product, total time).Workstation 3: Thermal welding machine 1 (see Figure 13c). The IR obstacle sensor KY-033 in this case detects the movement of the piston (when the piston is at the bottom, the machine is in the process of heat welding). The green pushbutton is used to terminate each product that has been made and the red pushbutton is used to enter the production order when there is a change. Finally, the keypad is used to enter the identification of the operator and the production order to be made. Using this node, the following data is collected: Production order that is being made; operator identification; time at which it is being carried out; production times (preparation and search time for aluminum profiles, heat-welding and winding time and total time).Workstation 4: Thermal welding machine 2 (see Figure 13d). In this workstation, IR obstacle sensor KY-033 has been used to detect the movement of the piston (when the piston is at the bottom, the machine is in the process of heat welding). The green pushbutton is used to terminate each product that has been made and the red pushbutton is used to enter the production order when there is a change. The 3 × 4 numeric Matrix Keypad is used to enter the identification of the operator and the production order to be made. The following data is collected: Production order that is being made; operator identification; time at which it is being carried out; production times (preparation time, heat-welding and preparation time and total time).Workstation 5: Automatic slat machine (see Figure 13e). The IR obstacle sensor KY-033 detects each time a slat is made, to obtain the times between each slat and to be able to count the number of slats made. The green pushbutton is used to terminate each product that has been made and the red pushbutton is used to enter the production order when there is a change. The Keypad is used to enter the identification of the operator and the production order to be made. The following data has been collected: Production order that is being made; operator identification; time in which it is being made; number of slats that have been made and the production times (programming and fabric search time, operating time, downtime and total time).

An important aspect of IIoT systems is the security of the communications. ZigBee mesh networks can employ robust security mechanisms, including 128-bit AES encryption and authentication protocols, to safeguard data integrity and prevent unauthorized access. Network key and link key encryption protect all transmitted data, while device commissioning and KEP ensure device legitimacy. Additional security measures like secure joining, ACLs, and message filtering further enhance network protection. The present study has been carried out without activating the encryption nor key use security mechanisms, following company’s requirements. In case this feature will be needed in the future, it can be easily configured. Its main effect on the system will be that the energy consumption of the nodes will grow. However, since all the nodes are connected to the power supply of the machines (i.e., to the power grid), no negative impact on the performance of the WSN will be expected.

## 4. Results and Discussion

In this context, the overall ZigBee-based WSN performance within the workshop has been assessed in terms of packet error rate (PER). As a first test, each ZigBee node has been programmed to send 10,000 packets in a time interval of 2 h, providing automatic link control supported by the ZigBee protocol. This measurement campaign has been performed for two different scenarios: during a high activity period at the workstations and during the rest period. Table 4 presents the correctly received packets by the coordinator from each node. It can be seen that no packet has been lost during both the activity and rest periods, obtaining a PER of 0.00% for both scenarios. These results confirm the results expected from the preliminary radio planning tasks presented in the previous section.

Then, the deployed WSN system has been tested for two complete working days (i.e., 48 h), during which the collected data has been stored and saved as an excel file. As an example, a macro of part of the information included in the stored packets is presented in Table 5, which shows the date, the workstation number, the worker ID, the fabrication order, and times (in seconds) related to specific tasks for each workstation.

The data collected from the implemented WSN has a lot of valuable information that can be integrated into the Enterprise Resource Planning (ERP) system for optimizing the production process, fabrication time, and enhancing the cost estimation of the products. In order to make this information available and accessible for those making decisions, several types of graphics have been implemented. An example of the extracted information from the WSN data is presented in Figure 14, where the involvement of employees during the fabrication of different products can be seen. Additionally, the different workstations participating in the process can be also seen, giving a complete and easy manufacturing overview. Further data analysis is presented in Figure 15, which shows: the time (in percentage) that each product manufacturing process consumed during the mentioned two complete working days (Figure 15a); how much time (in percentage) has spent a product in each workstation (Figure 15b); and the employees that took part in the manufacturing of a product (Figure 15c). Finally, Figure 16 presents the time spent manufacturing different products in the heat-welding workstation by two different employees.

As can be seen, the data collected from the deployed WSN, which is ready to be integrated into an ERP system, provides a wealth of information that can help the company optimize their production processes, adjust the working fluxes for more efficient performance, and enhance the cost estimation of products.

## 5. Conclusions

This paper presents a deployment of a low-cost ZigBee-based WSN in a real manufacturing facility. Galeo Enrollables Company, located in Navarre, Spain, manufactures technical and solar protection curtains, and it has bet in the Industry 4.0 paradigm by adding new capabilities to an ERP system. For that purpose, a ZigBee-based WSN has been deployed in order to monitor the five main manufacturing machines or workstations within their workshop and integrate them into an ERP system. The aim of the WSN is to acquire real-time data from the workstations and, thus, optimize the company’s production processes, enhance its cost estimation capabilities, improve the quality of their products, and make cost-reduction decisions. The wireless communication technology election was driven by the workshop’s morphology and size; and the radio channel and radio planning analysis presented in the paper showed that the WSN can successfully operate within this industrial environment. Interference levels within the factory are low, enabling the use of the proposed ZigBee network within the 2.4 GHz band. The proposed deterministic channel analysis supported by site spectral measurements can be generalized in the event that new types of interference sources could be included within the scenario (e.g., new types of machines that can include potentially interfering elements, such as soldering arches, electric brush motors, etc.), providing updated coverage/capacity analysis and hence, service provisions in the case of requiring higher bandwidth demands and/or an increased amount of sensors elements. The drawback of ZigBee technology could be the relative high-power consumption, but in this case all the nodes are connected to the workstation’s electric supply and no batteries are needed. Furthermore, ZigBee’s versatility (mesh topology, ease of adding new nodes to the network, …), WSN’s low cost (Arduino-compatible devices operating at free 2.4 GHz ISM band) and the obtained results led to satisfying the company’s expectations. For future work, the data collected by the WSN will be integrated into a new ERP system and analyses via Big Data techniques for the predictive optimization of the manufacturing processes will be performed. Moreover, other aspects, such as security, should be considered in a security by design approach in order to consider requirements in terms of internal wireless connectivity (updates in relation to the currently supported AES 128-bit ZigBee scheme) as well as external connectivity at the edge/cloud.

## Figures and Tables

**Figure 1 sensors-24-00712-f001:**
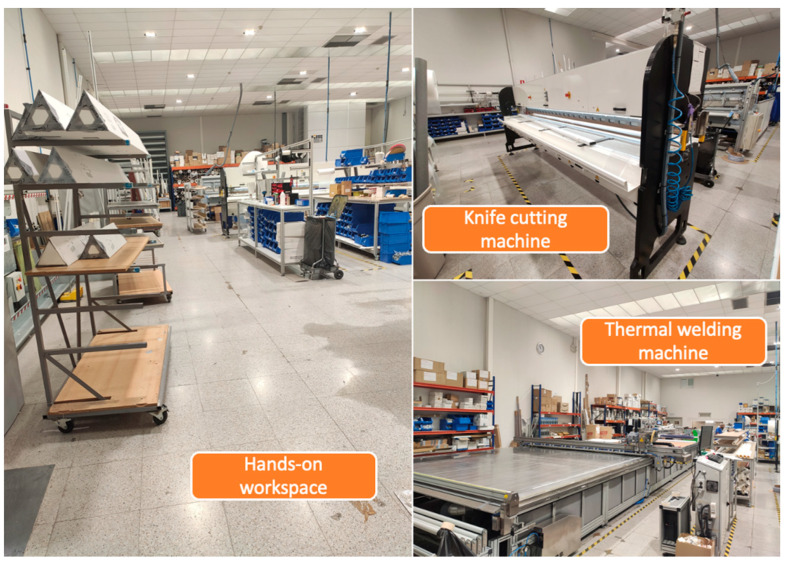
Different workstations and areas of the Galeo factory workshop.

**Figure 2 sensors-24-00712-f002:**
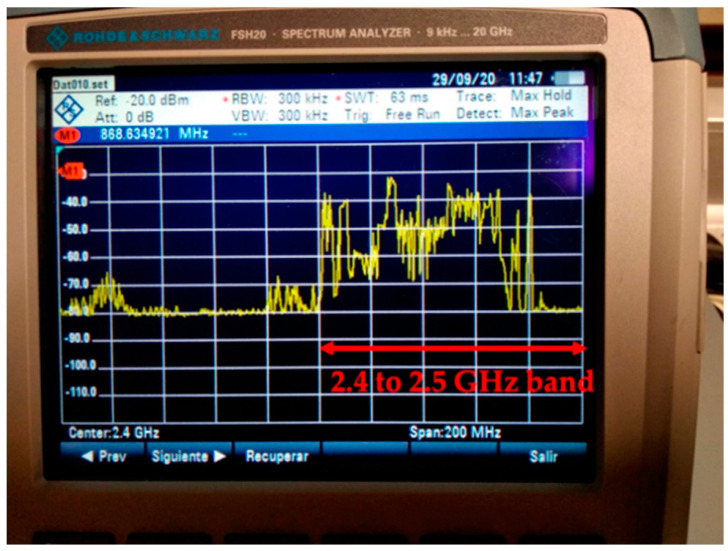
RF interference measurements within the factory workshop at the 2.4 GHz band.

**Figure 3 sensors-24-00712-f003:**
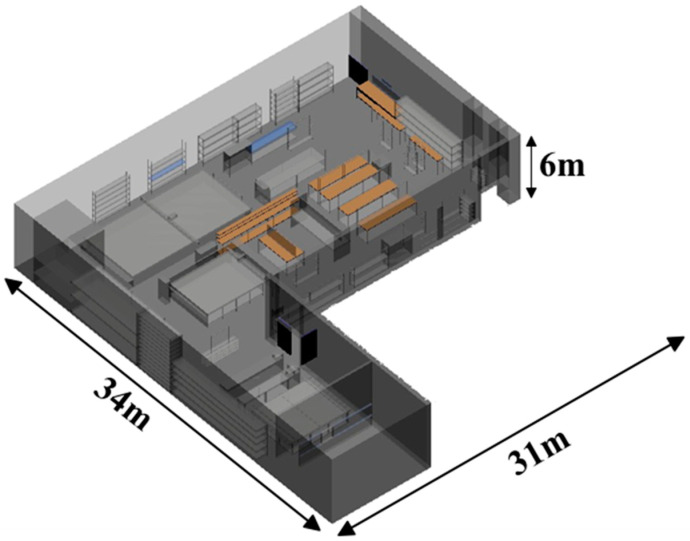
The created workshop scenario for its simulation by the 3D Ray Launching tool.

**Figure 4 sensors-24-00712-f004:**
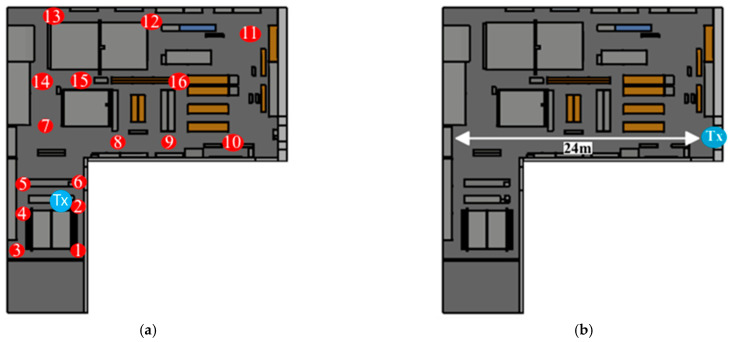
The two measurement campaigns carried out for the validations of the 3D ray launching algorithm, (**a**) measurement points (1 to 16) covering the entire area of the workshop and (**b**) measurements for a LoS linear path.

**Figure 5 sensors-24-00712-f005:**
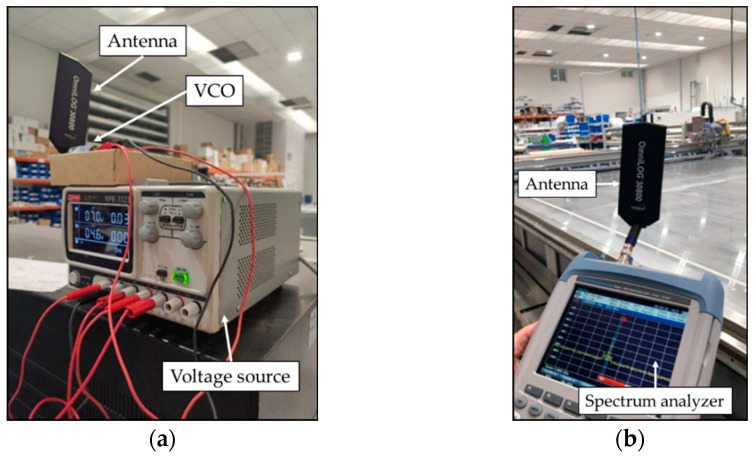
Employed transmitter and receiver configuration for the radio channel measurements: (**a**) the VCO as a transmitter; (**b**) the spectrum analyzer as a receiver.

**Figure 6 sensors-24-00712-f006:**
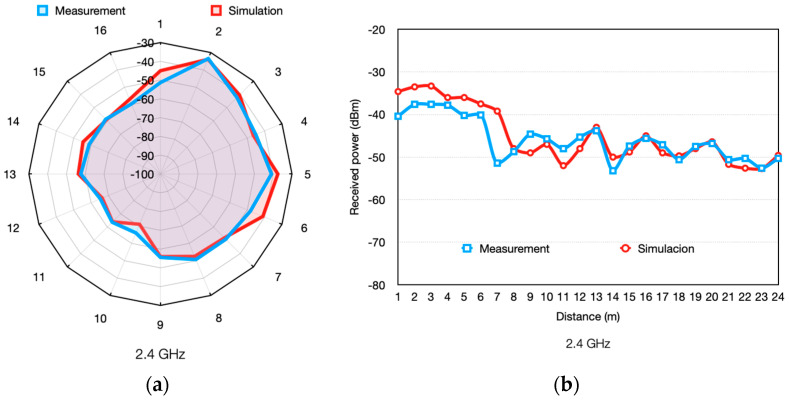
Measurements vs. 3D-RL simulation results for (**a**) scenario 1 and (**b**) scenario 2.

**Figure 7 sensors-24-00712-f007:**
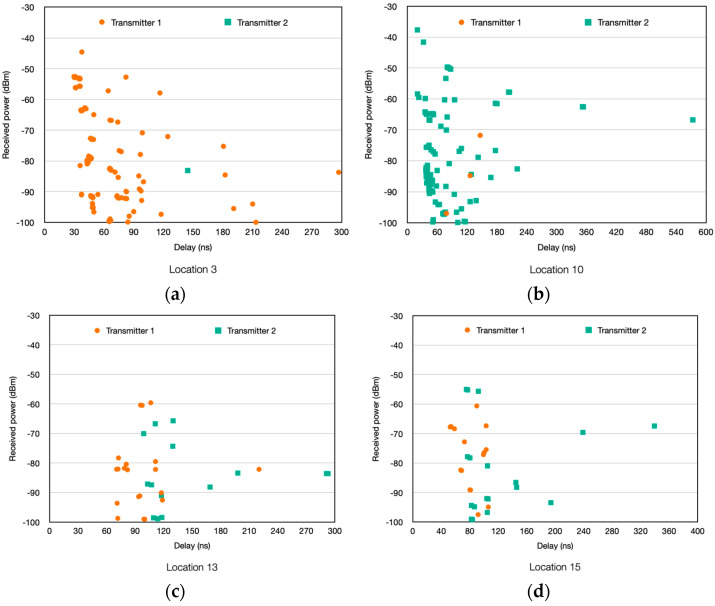
Estimated power delay profiles at different locations (measurement points) for both TX1 and TX2 (**a**) location 3; (**b**) location 10; (**c**) location 13; (**d**) location 15.

**Figure 8 sensors-24-00712-f008:**
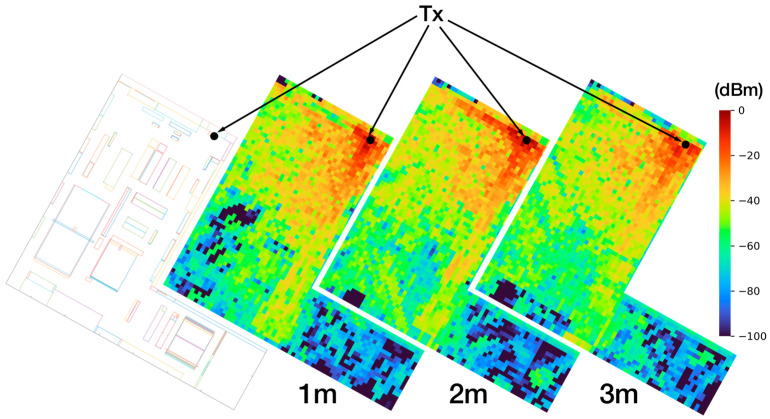
The diagram shows 2.4 GHz RF power level distribution maps at different heights for scenario 2.

**Figure 9 sensors-24-00712-f009:**
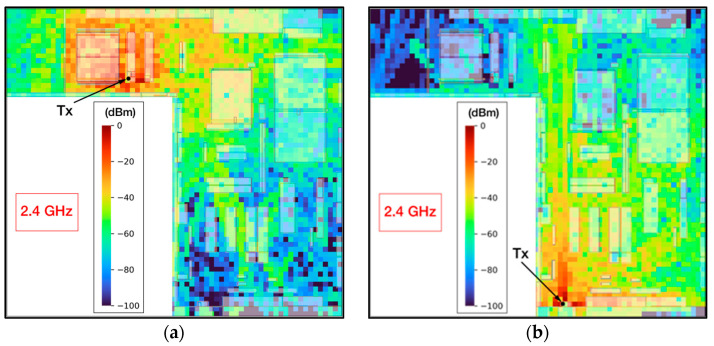
Bidimensional RF power level distribution planes for (**a**) scenario 1, and (**b**) scenario 2.

**Figure 10 sensors-24-00712-f010:**
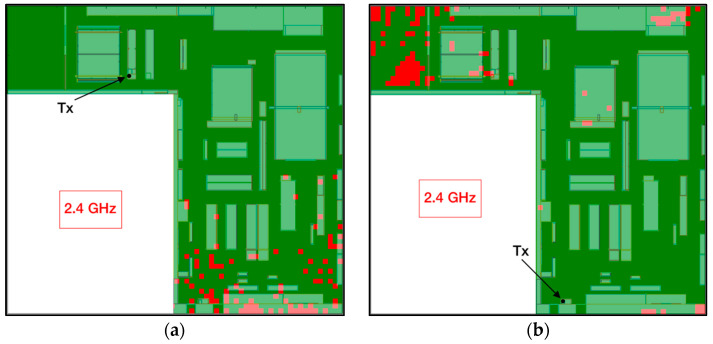
Two-dimensional planes for ZigBee notes’ sensitivity compliance for (**a**) scenario 1 and (**b**) scenario 2.

**Figure 11 sensors-24-00712-f011:**
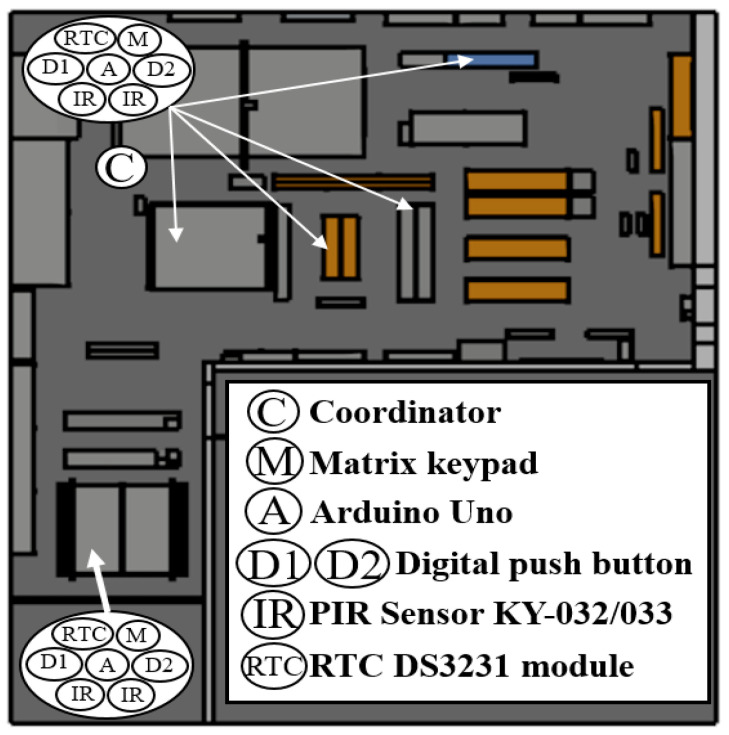
Schematic view of the WSN deployment within the workshop (‘C’ represents the ZigBee network coordinator/gateway).

**Figure 12 sensors-24-00712-f012:**
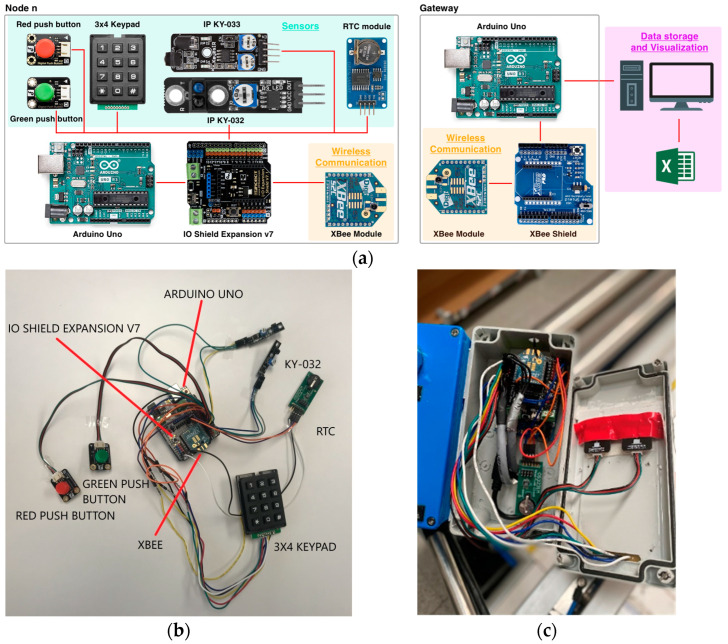
(**a**) The implemented sensor nodes and Coordinator/Gateway; (**b**) Picture of a sensor node; (**c**) The node encapsulated.

**Figure 13 sensors-24-00712-f013:**
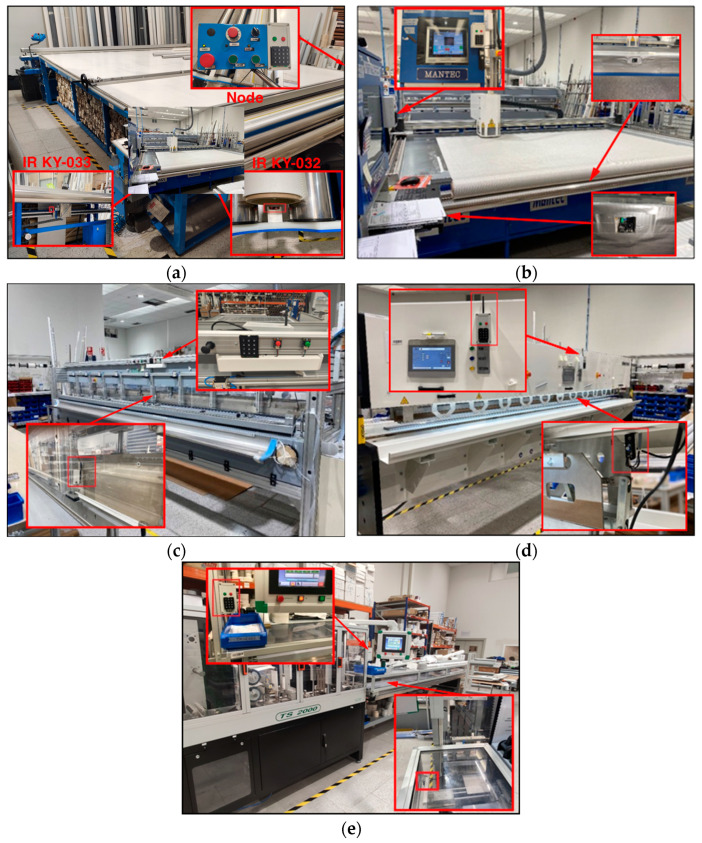
The implemented nodes at different workstations: (**a**) knife cutting machine; (**b**) laser cutting machine; (**c**) thermal welding machine 1; (**d**) thermal welding machine 2; (**e**) automatic slat machine.

**Figure 14 sensors-24-00712-f014:**
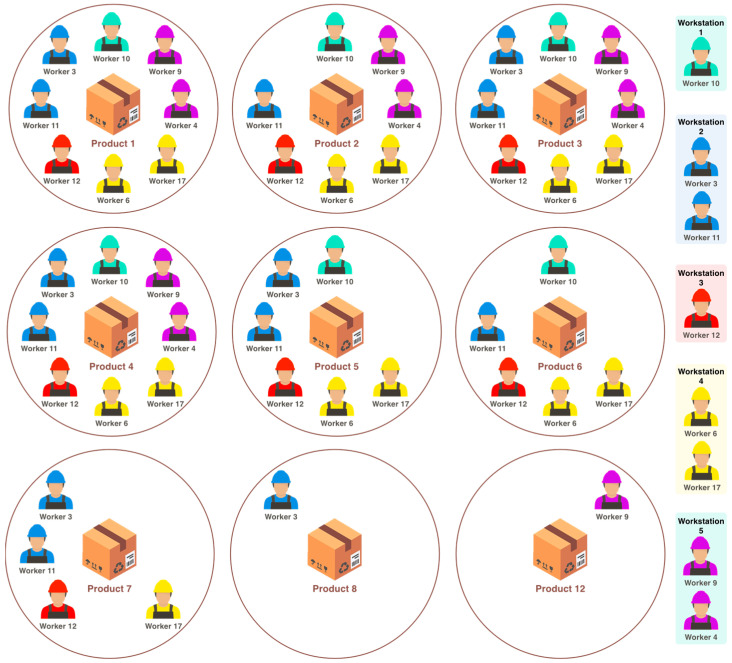
Participation of employees and workstations in different product fabrications.

**Figure 15 sensors-24-00712-f015:**
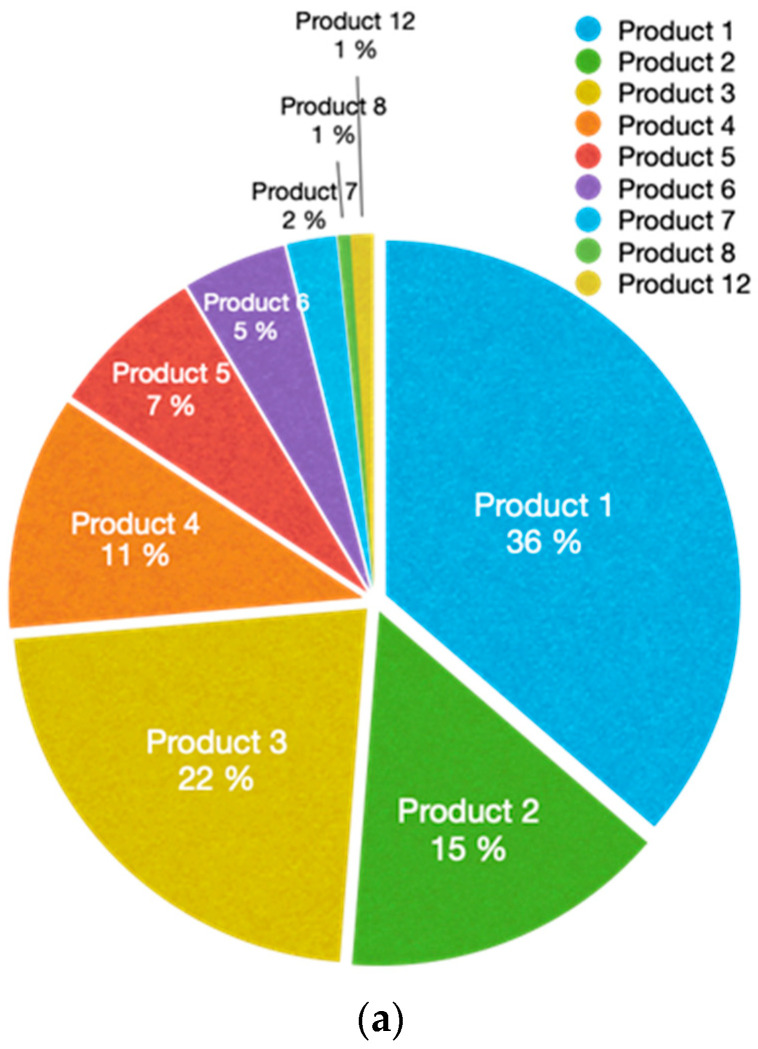
Data analytics examples: (**a**) time that each product manufacturing process consumed during two complete working days; (**b**) time spent by a product at each workstation; (**c**) employees that took part in the manufacturing of a product.

**Figure 16 sensors-24-00712-f016:**
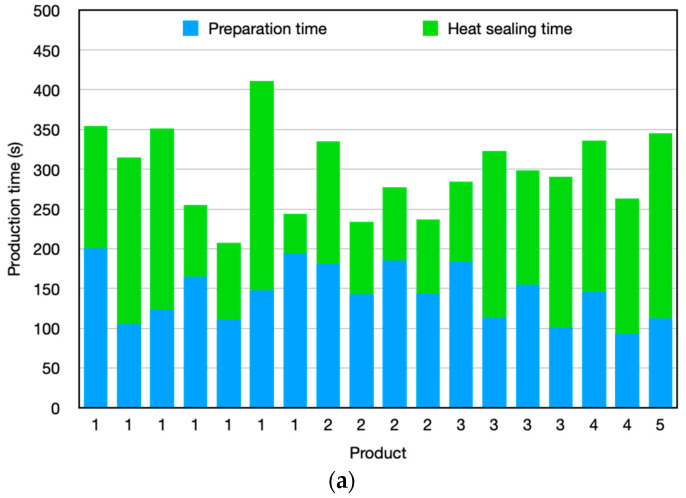
Time consumed manufacturing products in the heat-welding workstation, by two different workers: (**a**) Employee 6; (**b**) Employee 17.

**Table 1 sensors-24-00712-t001:** Relative permittivity and conductivity of the existing elements within the created scenario.

Elements	$\mathrm{Relative}\;\mathrm{Permittivity}\;({ԑ}_{r})$	Conductivity (σ) [S/m]
Air	1	0
Metal	4.5	37.8 × 10^6^
Glass	6.06	0.11
Plastic	8.5	0.02
PVC	4	0.12
Brick wall	4.44	0.11

**Table 2 sensors-24-00712-t002:** Parameter configuration for the simulations using the 3D-RL tool.

Parameter	Value
Operating frequency	2.4 GHz
Transmitted power	8.2 dBm
TX gain	−6 dBi
TX1/TX2 heights	1.75 m/1.2 m
Launched ray resolution (ΔΦ)	1°
Maximum number of reflections	6
Mesh resolution	50 cm × 50 cm × 20 cm

**Table 3 sensors-24-00712-t003:** Obtained mean error, variance, and standard deviation for the measurement-simulation comparison.

Scenario	Mean Error (dB)	Variance	Standard Deviation (σ)
1	2.38	4.36	2.08
2	2.58	6.46	2.54

**Table 4 sensors-24-00712-t004:** Correctly received packets by the coordinator from each WSN node (from 10,000 transmitted packets) during activity and rest periods.

Node	Activity Period	Rest Period
Cutting table	10,000	10,000
Laser	10,000	10,000
Soldering	10,000	10,000
Automatic slats	10,000	10,000

**Table 5 sensors-24-00712-t005:** Raw data with part of the information sent in the packets.

Time	Date	Workstation	Worker	Fab. Order	T1 ID	T1 Time	T2 ID	T2 Time
14:01:27	7 October 2020	1	34	0	1			
14:01:44	7 October 2020	1	34	78,055	8	17		
14:03:00	7 October 2020	1	34	78,055	9	0	9	19
14:04:55	7 October 2020	1	34	78,055	10	39	15	24
14:06:50	7 October 2020	1	34	78,055	11	52	9	23

## Data Availability

Data are contained within the article.

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
