# Peer review of "IIoT Low-Cost ZigBee-Based WSN Implementation for Enhanced Production Efficiency in a Solar Protection Curtains Manufacturing Workshopâ€"

_sensors, 2024, doi:10.3390/s24020712_

Round 1
Reviewer 1 Report
Comments and Suggestions for Authors
1. Title of the work has not been reflected in abstract
2. transmitter power is not convincing and gain is not validated with the design concept
3. there is no consistency in the methodology adopted
4. deployment of the network should have been validated with the conceptual communication standards
5. results are not validated and need more clarity and convincing
Comments on the Quality of English Language
it is requiring minor updation
Reviewer 2 Report
Comments and Suggestions for Authors
sensors-2793327
Title: IIoT Low-cost ZigBee-based WSN implementation for Enhanced Production Efficiency in a Solar Protection Curtains Manufacturing Workshop
Some points need to be known.
- Figure 1 should be labeled appropriately.
- What are the limitations of the Zigbee protocol? Can someone use some other protocol in place of ZIgbee e.g. LoRA etc?
- In the mesh network, security is crucial. How do you use encryption and proper authentication methods to protect your network?
-What are the proposed Zigbee mesh network's Latency, scalability, and power consumption?
-Figure 2 shows that there is some RF interference within the factory workshop at the 2.4 GHz band, but the Packet Error Rate (PER) is 0% (line 430). How is it possible?
-Can someone use ESP mesh network in place of Zigbee mesh network as used in https://doi.org/10.3390/su142416630
Comments on the Quality of English LanguageMinor editing of the English language is required.
Reviewer 3 Report
Comments and Suggestions for Authors
Suggestions for Minor Revision:
1. Detailed Analysis of Electromagnetic Interference: While the paper discusses the challenges of electromagnetic noise, a more comprehensive analysis of potential interferences, especially from specific machinery, could strengthen the paper.
2. Long-term Performance Evaluation: The paper presents initial performance results, but a long-term evaluation would provide more insights into the reliability and scalability of the WSN over time.
3. Energy Efficiency Analysis: Given the concern about ZigBee's power consumption, an in-depth analysis of the energy efficiency of the WSN could be beneficial.
4. Broader Contextualization Within Industry 4.0: While the paper aligns with Industry 4.0, providing a broader context of how this implementation fits into the global trends in industrial automation could offer more depth.
5. Data Security Considerations: As the WSN deals with real-time production data, addressing data security and privacy concerns would be prudent.
These suggestions aim to enhance the comprehensiveness and applicability of the research, ensuring that the study not only presents a successful implementation but also provides a thorough analysis and consideration of all relevant aspects of WSN deployment in an industrial setting.
Round 2
Reviewer 1 Report
Comments and Suggestions for Authors
1. abstract still not compatible to the contents
2. Link formation is not explained well
3. How energy is saved by having 6 reflection
4. how RF power is managed in the area is not seen in the work
5. results are not validated
Comments on the Quality of English Languageneed check language of abstract and result section
Reviewer 2 Report
Comments and Suggestions for Authors
sensors-2793327
IIoT Low-cost ZigBee-based WSN implementation for Enhanced Production Efficiency in a Solar Protection Curtains Manufacturing Workshop
Thank you for allowing me to revise the resubmitted manuscript titled " IIoT Low-cost ZigBee-based WSN implementation for Enhanced Production Efficiency in a Solar Protection Curtains Manufacturing Workshop." I believe the submitted manuscript and presented work is suitable for publishing in the Sensors.
